# Peer review of "Supply chain resource scheduling optimization of e-commerce enterprises in international trade based on mobile edge computing"

_PeerJ Computer Science, doi:10.7717/peerj-cs.1407_

## Round 0.1 · original submission · Minor Revisions

Please revise your paper carefully according to expert's comments and resubmit.

Reviewer 1 ·

Basic reporting

In this study, a genetic algorithm constructed and solved the resource scheduling model of the supply chain of e-commerce enterprises in international trade. In addition, it involves MEC optimization based on partial computing offloading, setting the initial offloading ratio and allocating supply chain resources, then allocating remaining computing resources based on server computing capacity, and optimizing offloading based on resource allocation. In summary, the supply chain optimization scheme proposed in this paper can effectively utilize supply chain resources according to the requirements of computing tasks to reduce the total delay of task execution and the consumption of node computing. The following are specific suggestions for modification:
(1) "Cost Optimization Control" in the title is not reflected in the experimental results, which is unreasonable;
(2) Please note that acronyms of terms used just once in the abstract need not be included. Instead, the acronyms can be introduced in the main text, where they are repeatedly mentioned;
(3) The optimization model with the total cost and total time as the objective function seems to be very common, and Lines 89-92 need to be modified;
(4) Some contents are missing, and more descriptions are added to formulas (3) ~ (6);
(5) What are the implementation steps of the linear weighting method? How it solves the proposed model;
(6) Resource calculation after user update is a dynamic calculation process, and there seems to be no discussion of the unloading ratio in the result analysis;
(7) In the result analysis, it is necessary to add a paragraph to discuss the application scenarios of the model;
(8) Literature Review has the chance to be further improved: it seems the authors have made the retrospection. However, via the review, what issues should be addressed?
(9) What is the current specific knowledge gap? What implication can be referred to? The above questions should be answered point-by-point.

Experimental design

In this study, a genetic algorithm constructed and solved the resource scheduling model of the supply chain of e-commerce enterprises in international trade. In addition, it involves MEC optimization based on partial computing offloading, setting the initial offloading ratio and allocating supply chain resources, then allocating remaining computing resources based on server computing capacity, and optimizing offloading based on resource allocation. In summary, the supply chain optimization scheme proposed in this paper can effectively utilize supply chain resources according to the requirements of computing tasks to reduce the total delay of task execution and the consumption of node computing. The following are specific suggestions for modification:
(1) "Cost Optimization Control" in the title is not reflected in the experimental results, which is unreasonable;
(2) Please note that acronyms of terms used just once in the abstract need not be included. Instead, the acronyms can be introduced in the main text, where they are repeatedly mentioned;
(3) The optimization model with the total cost and total time as the objective function seems to be very common, and Lines 89-92 need to be modified;
(4) Some contents are missing, and more descriptions are added to formulas (3) ~ (6);
(5) What are the implementation steps of the linear weighting method? How it solves the proposed model;
(6) Resource calculation after user update is a dynamic calculation process, and there seems to be no discussion of the unloading ratio in the result analysis;
(7) In the result analysis, it is necessary to add a paragraph to discuss the application scenarios of the model;
(8) Literature Review has the chance to be further improved: it seems the authors have made the retrospection. However, via the review, what issues should be addressed?
(9) What is the current specific knowledge gap? What implication can be referred to? The above questions should be answered point-by-point.

Validity of the findings

In this study, a genetic algorithm constructed and solved the resource scheduling model of the supply chain of e-commerce enterprises in international trade. In addition, it involves MEC optimization based on partial computing offloading, setting the initial offloading ratio and allocating supply chain resources, then allocating remaining computing resources based on server computing capacity, and optimizing offloading based on resource allocation. In summary, the supply chain optimization scheme proposed in this paper can effectively utilize supply chain resources according to the requirements of computing tasks to reduce the total delay of task execution and the consumption of node computing. The following are specific suggestions for modification:
(1) "Cost Optimization Control" in the title is not reflected in the experimental results, which is unreasonable;
(2) Please note that acronyms of terms used just once in the abstract need not be included. Instead, the acronyms can be introduced in the main text, where they are repeatedly mentioned;
(3) The optimization model with the total cost and total time as the objective function seems to be very common, and Lines 89-92 need to be modified;
(4) Some contents are missing, and more descriptions are added to formulas (3) ~ (6);
(5) What are the implementation steps of the linear weighting method? How it solves the proposed model;
(6) Resource calculation after user update is a dynamic calculation process, and there seems to be no discussion of the unloading ratio in the result analysis;
(7) In the result analysis, it is necessary to add a paragraph to discuss the application scenarios of the model;
(8) Literature Review has the chance to be further improved: it seems the authors have made the retrospection. However, via the review, what issues should be addressed?
(9) What is the current specific knowledge gap? What implication can be referred to? The above questions should be answered point-by-point.

Additional comments

In this study, a genetic algorithm constructed and solved the resource scheduling model of the supply chain of e-commerce enterprises in international trade. In addition, it involves MEC optimization based on partial computing offloading, setting the initial offloading ratio and allocating supply chain resources, then allocating remaining computing resources based on server computing capacity, and optimizing offloading based on resource allocation. In summary, the supply chain optimization scheme proposed in this paper can effectively utilize supply chain resources according to the requirements of computing tasks to reduce the total delay of task execution and the consumption of node computing. The following are specific suggestions for modification:
(1) "Cost Optimization Control" in the title is not reflected in the experimental results, which is unreasonable;
(2) Please note that acronyms of terms used just once in the abstract need not be included. Instead, the acronyms can be introduced in the main text, where they are repeatedly mentioned;
(3) The optimization model with the total cost and total time as the objective function seems to be very common, and Lines 89-92 need to be modified;
(4) Some contents are missing, and more descriptions are added to formulas (3) ~ (6);
(5) What are the implementation steps of the linear weighting method? How it solves the proposed model;
(6) Resource calculation after user update is a dynamic calculation process, and there seems to be no discussion of the unloading ratio in the result analysis;
(7) In the result analysis, it is necessary to add a paragraph to discuss the application scenarios of the model;
(8) Literature Review has the chance to be further improved: it seems the authors have made the retrospection. However, via the review, what issues should be addressed?
(9) What is the current specific knowledge gap? What implication can be referred to? The above questions should be answered point-by-point.

Reviewer 2 ·

Basic reporting

Based on the research of CBESCN in various aspects, an optimization model with CBESCN characteristics is constructed in the corresponding mode. Through the joint allocation of computing resources of node devices, supply chain resources, and computing resources of edge servers, a method is proposed to unload the computation-intensive tasks of nodes to edge servers. Considering the allocation of server computing resources, the average delay of the proposed method is low. Although it is helpful to provide new ideas for the theoretical development of CBESCN and its service supply chain.
1- "CBE" needs to be accurately defined to complement its full name;
2- Please replace the references in literature 17 and 24 and supplement the English literature; For example: Li R, Li Q, Zhou J, et al. ADRIoT: an edge-assisted anomaly detection framework against IoT-based network attacks[J]. IEEE Internet of Things Journal, 2021, 9(13): 10576-10587;

Experimental design

3-The design of the model needs to consider spanning two or more different countries/regions.
4-The research in this paper focuses on supply chain optimization in international trade, but I haven't seen more background introduction on the international scope.
5-Line 181, the sentence ” The genetic algorithm in MATLAB is used to solve the objective function.” is confusing.

Validity of the findings

6- Why "the calculation result of the objective function K is equal to 0.6612." Add more data to discuss this.
7-The description and analysis of Figure 3 are too lengthy.
8-Focus on the logical correlation between data analysis. "However, this improvement is not infinite."
9-The actual value of this work is not significant at present. How can it achieve the purpose of inventory optimization as mentioned in the title?

---

## Round 0.2 · accepted · Accept

The reviewers are satisfied with the revision; I'm pleased to inform you about the acceptance of your article. Thanks for your contribution to our journal

Reviewer 1 ·

Basic reporting

The revised paper is good.
The requirements are done.

Experimental design

The revised paper is good.
The requirements are done.

Validity of the findings

The revised paper is good.
The requirements are done.

Additional comments

The revised paper is good.
The requirements are done.

Reviewer 2 ·

Basic reporting

I have seen the revision according to the previous comments and it seems good now.

Experimental design

In good condition after the updated version.

Validity of the findings

The improvements are upto the requirement and the paper is in condition to be accepted.